

**Averaging over spatiotemporal heterogeneity substantially biases evapotranspiration rates in a mechanistic**
**large-scale land evaporation model**
Elham Rouholahnejad Freund[1,2,3], Massimiliano Zappa[4], James W. Kirchner[3,4,5]
[1]Laboratory of Hydrology and Water Management, Ghent University, Ghent, Belgium
[2]Chair of Hydrology, Faculty of Environment and Natural Resources, University of Freiburg, Freiburg, Germany
[3]Department of Environmental Systems Science, ETH Zurich, CH-8092 Zürich, Switzerland
[4]Swiss Federal Research Institute WSL, CH-8903 Birmensdorf, Switzerland
[5]Department of Earth and Planetary Science, University of California, Berkeley, CA 94720 USA
Correspondence to: Elham Rouholahnejad Freund, elham.rouholahnejad@gmail.com
**Abstract**
Evapotranspiration (ET) influences land-climate interactions, regulates the hydrological cycle, and contributes
to the Earth's energy balance. Due to its feedbacks to large-scale hydrological processes and its impact on
atmospheric dynamics, ET is a key driver of droughts and heatwaves. Existing land surface models differ
substantially, both in their estimates of current ET fluxes and in their projections of how ET will evolve in the
future. Any bias in estimated ET fluxes will affect the partitioning between sensible and latent heat, and thus
alter model predictions of temperature and precipitation. One potential source of bias is the so-called
"aggregation bias" that arises whenever nonlinear processes, such as those that regulate ET fluxes, are
modeled using averages of heterogeneous inputs. Here we demonstrate a general mathematical approach to
quantifying and correcting for this aggregation bias, using the GLEAM land evaporation model as a relatively
simple example. We demonstrate that this aggregation bias can lead to substantial overestimates in ET fluxes
in a typical large-scale land surface model when sub-grid heterogeneities in land surface properties are
averaged out. Using Switzerland as a test case, we examine the scale-dependence of this aggregation bias and
show that it can lead to overestimation of daily ET fluxes by as much as 21% averaged over the whole country.
We show how our approach can be used to identify the dominant drivers of aggregation bias, and to estimate
sub-grid closure relationships that can correct for aggregation biases in ET estimates, without explicitly
representing sub-grid heterogeneities in large-scale land surface models.
**Plain Language Summary**
Evapotranspiration (ET) is the largest flux from the land to the atmosphere and thus contributes to Earth's
energy and water balance. Due to its impact on atmospheric dynamics, ET is a key driver of droughts and
heatwaves. In this paper, we demonstrate how averaging over land surface heterogeneity contributes to
substantial overestimates of ET fluxes. We also demonstrate how one can correct for the effects of small-scale
heterogeneity without explicitly representing it in land surface models.






**1.   Introduction**

Earth's surface and subsurface are characterized by spatial heterogeneity spanning wide ranges of scales,
including scales that cannot be explicitly resolved by large-scale Earth System Models (ESMs), which are
typically run at resolutions of 10-100 kilometers. Averaging over this finer-scale heterogeneity can bias model
estimates of water and energy fluxes and hence alter future temperature predictions. Earth system model
estimates of global terrestrial evaporation differ substantially from atmospheric reanalyses based on in-situ
and satellite remote sensing observations (Mueller et al., 2013), but it is unclear how much of these
differences could be attributed to errors in capturing sub-grid heterogeneity.

Several recent studies (e.g., Fan et al., 2019; Shrestha et al., 2018) have emphasized the need to  account for
land surface heterogeneity in large-scale ESMs. Despite recent community efforts in refining ESMs' spatial
resolution (Huang et al., 2016; Rauscher et al., 2010; Ringler et al., 2008; Skamarock et al., 2012; Zarzycki et al.,
2014), the grid resolution of present-day ESMs is still too coarse to explicitly capture important effects of
surface heterogeneity. Whether the solution lies in hyper-resolution large-scale land surface modeling remains
an open question, because heterogeneities that are important to land-atmosphere fluxes will not be fully
resolved even at scales of 100 m (Beven and Cloke, 2012).

The effects of aggregating over spatial heterogeneity in land surface models have been assessed using several
approaches. Most of these approaches compare grid-cell-averaged energy and water fluxes with flux estimates
for finer-resolution grids, or for grid cells that are subdivided into mosaics of several surface types which
separately exchange momentum, energy, and water vapor with the overlying atmosphere (e.g., Giorgi, 1997).
Several studies have reported increases in average evapotranspiration (ET), and at least one has reported
decreases in grid-cell average ET, as model grids are coarsened and less spatial heterogeneity is accounted for
(e.g., Kuo et al., 1999; Boone and Wetzel, 1998; Hong et al., 2009; McCabe and Wood, 2006; Ershadi 2013; El
Maayar and Chen, 2006). Shrestha et al. (2018) studied the effects of horizontal grid resolution on ET
partitioning in the TerrSysMP Earth system model and found that the aggregation of topography decreases
average slope gradients and obscures small-scale convergence and divergence zones, directly impacting
surface and subsurface flow. They observed 5 and 8 percent decreases in the transpiration/evapotranspiration
ratio for a dry and a wet year, respectively, when their model grid cells were coarsened from 120 m to 960 m.
All these studies calculate the effects of land surface heterogeneity on ET fluxes using numerical experiments
that refine the model's spatial resolution, either directly or through the use of land-surface mosaics.

Quantifying the effect of sub-grid scale heterogeneity on grid-cell-averaged fluxes is especially important when
highly nonlinear processes are involved. Regardless of scale, the main challenge is not to explicitly represent
the heterogeneity in all its details, but instead to define an appropriate scale-dependent sub-grid closure
relationship that recognizes the important heterogeneities within the grid elements and the nonlinearities in



the processes (Beven, 2006). Such a sub-grid closure scheme would capture the effects of sub-grid
heterogeneity in large-scale land surface models without forcing them to run at finer spatial resolutions.

We have recently proposed a general theoretical framework, based on Taylor series expansions, that
quantifies the "aggregation bias" that results from averaging over sub-grid heterogeneity when grid-cell-
averaged ET is estimated (Rouholahnejad Freund and Kirchner, 2017; Rouholahnejad Freund et al., 2019). In
contrast to the numerical experiments described above, this theoretical framework does not depend on a
particular evapotranspiration model or grid scale. Our previous work demonstrated this framework using
Budyko curves as a see-through "toy" model, leaving open the question of how strongly ET estimates would be
affected by sub-grid heterogeneity in a more typical mechanistic evapotranspiration model. Here we use the
mechanistic evapotranspiration model GLEAM to quantify how aggregation biases vary across a range of
scales, using Switzerland as a case study. We show how our Taylor expansion framework can be used to
quantify the sensitivity of ET fluxes to heterogeneity in their individual drivers. We further demonstrate how
this framework can be used to estimate correction factors (i.e., sub-grid closure relationships) that account for
the effects of sub-grid heterogeneity without explicitly modeling it, and show how these correction factors can
be used to improve grid-scale ET estimates. Because our framework is not model-specific, the analysis
presented here could also be applied to many other evapotranspiration algorithms.

**2.   Methods and results**
**2.1. A common mechanistic framework for predicting evapotranspiration**
Most large-scale land surface models calculate ET as a function of available water and energy at daily time
steps. They typically multiply an estimate of potential evapotranspiration (PET) by a conversion factor to
calculate actual evapotranspiration. PET is generally understood as the maximum rate of evapotranspiration
from a large area (to avoid the effect of local advection) covered completely and uniformly by actively growing
vegetation with adequate moisture at all times (Brutseart, 1984). Models typically estimate PET using the
Penman equation (Penman, 1948; intended for open water surfaces), the Penman-Monteith equation
(Monteith, 1965, Monteith and Unsworth, 1990; intended for reference crop evapotranspiration by adding
atmospheric transport processes and stomatal resistance to Penman's open water evaporation), or the
Priestley-Taylor equation (Priestley and Taylor, 1972; intended for open water and water-saturated crops and
grasslands). The conversion factor that is used to estimate ET from PET typically depends on plant physiology
and on the water that is available for evaporation.

Here, we employ an ET algorithm that is used by several land surface models (i.e., Global Land-surface
Evaporation: The Amsterdam Methodology (GLEAM); Miralles et al., 2011; Martens et al., 2017), in which
actual ET is calculated as a fraction of PET. This fraction is expressed as a multiplicative factor, often called a
stress factor, which ranges between 0 and 1 and thus limits ET rates. Under wet conditions, ET can equal PET
(stress factor equals one) while under dry conditions, PET is multiplied by a stress factor smaller than one
depending on the degree of water stress. This approach is employed by the GLEAM model, among others.





GLEAM is a diagnostic satellite-data-driven method that is used to estimate global land evaporation fluxes.
GLEAM uses the Priestley-Taylor formula and remotely sensed datasets of radiation and temperature to
calculate PET. In GLEAM, actual ET is calculated by constraining PET estimates by a stress factor that is based
on estimates of root-zone soil moisture. The root zone soil moisture is derived from a multi-layer water
balance module that describes the infiltration of precipitation through the vertical soil profile. ET estimates
from GLEAM have been applied in many studies (e.g., Miralles et al., 2013; Miralles et al., 2014; Greve et al.,
2014; Jasechko et al., 2013). GLEAM operates on daily time steps at 0.25-degree spatial resolution. To the best
of our knowledge, there are no prior studies quantifying the aggregation bias in ET estimates from GLEAM or
other models with similar ET formulations.

GLEAM calculates ET as an explicit function of the stress factor and potential evaporation:
$$ET = S \cdot PET + (1 - \beta)\,I,\tag{1}$$
where $ET$ is actual evapotranspiration (mm d$^{-1}$), $S$ is the evaporative stress factor (-) that accounts for
environmental conditions that reduce actual ET relative to potential ET, $I$ is interception losses (mm d$^{-1}$), and $\beta$
is a constant ($\beta$= 0.07 – Gash and Stewart, 1977) that avoids double-counting of interception losses during
hours with wet canopy. The stress factor ($S$) depends on the soil moisture conditions, and is parametrized
separately for tall canopy, short vegetation, and bare soil. GLEAM uses the following soil-moisture-based
parameterization to calculate the stress factor (Miralles et al., 2011; Martens et al., 2017):
$$S = 1 - \left(\frac{w_c - w_w}{w_c - w_{wp}}\right)^2,\tag{2}$$
where $S$ is the stress factor (-) for tall canopy, $w_w$ is the volumetric soil moisture at any given time (m$^3$ m$^{-3}$), and
$w_c$ and $w_{wp}$ are the critical soil moisture level and soil moisture at wilting point. For soil moisture values below
the wilting point $w_{wp}$, the stress is maximal (stress factor equals 0), causing ET to sharply decline to zero. For
values above the critical moisture level $w_c$, there is no water stress (stress factor equals 1) and ET equals PET.
Between $w_{wp}$ and $w_c$ the stress increases as soil moisture decreases following a parabolic function (Eq. 2). In
the analysis presented below, we set the critical soil moisture level ($w_c$) and soil moisture at wilting point
($w_{wp}$) to 0.6 and 0.1 m$^3$ m$^{-3}$ respectively. To simplify the analysis presented below, we have used the tall-
canopy stress factor (Eq. 2) for all of Switzerland, even though the short-canopy or bare-soil formulations may
be better suited to some locations.

GLEAM uses the Priestley-Taylor approach to calculate PET (Priestley and Taylor, 1972):
$$PET = \frac{\alpha}{\lambda}\frac{\Delta}{\Delta + \gamma}(R_n - G),\tag{3}$$
where $PET$ is potential evapotranspiration (mm d$^{-1}$), $\alpha$ is a dimensionless coefficient that parametrizes the
resistance to evaporation and is set to 0.8 for tall canopy in GLEAM (Miralles et al., 2011), $\lambda = 2.26$ (MJ kg$^{-1}$) is
the latent heat of vaporization, $R_n$ is net radiation (MJ m$^{-2}$ d$^{-1}$), $G$ is the ground heat flux, approximated as
$G$=0.05 $R_n$ (MJ m$^{-2}$ d$^{-1}$) for tall canopy in GLEAM, $T$ is temperature (°C), and $\Delta$ is the slope of the



temperature/saturated vapor pressure curve (kPa°C⁻¹), which is functionally related to temperature (Tetens,
1930; Murray, 1967; Stanghellini, 1987):
$$\Delta = ae^{bT},\qquad(4)$$
where $a$= 0.04145 (kPa°C⁻¹), $b$=0.06088 (°C⁻¹), and $\gamma$ is the psychrometric constant (kPa°C⁻¹) which can be
calculated as (Brunt, 1952):
$$\gamma = \frac{C_{p_{air}} * P}{\lambda * MW_{ratio}},\qquad(5)$$
where $C_{p_{air}}$ = 0.001013 (MJ kg⁻¹°C⁻¹) is the specific heat of air at constant pressure, $P$ = 101.3 (KPa) is
atmospheric pressure, and $MW_{ratio}$ =0.622 (-) is the molecular weight ratio of $H_2O$/air. Substituting the
aforementioned constants in Eq. 5 yields $\gamma$= 0.073 (kPa°C⁻¹). Expanding Eq. 1 using Eqs. 2-5 yields the ET
function as calculated by GLEAM:

$$ET_{[mmd^{-1}]} = \left[-4w_{w[m^3m^{-3}]}^2 + 4.8w_{w[m^3m^{-3}]} - 0.44\right] * \frac{\alpha_{[\ ]}}{\lambda_{[MJ\ kg^{-1}]}} * \frac{\Delta_{[kPa°C^{-1}]}}{\Delta_{[kPa°C^{-1}]} + \gamma_{[kPa°C^{-1}]}}$$

$$* 0.95 * \frac{86400}{1000000} * R_{n[Wm^{-2}]} + (1 - \beta)\, I_{[mmd^{-1}]}$$

$$= [-4w_w^2 + 4.8w_w - 0.44] * 0.02905 * \frac{a\, e^{bT}}{a\, e^{bT} + 0.073} R_n + (1 - 0.07)\, I_{[mmd^{-1}]}.\qquad(6)$$


In the analysis below, we use GLEAM to demonstrate how aggregation biases can be estimated in land surface
modeling schemes. We chose GLEAM because its governing equations are amenable to the analytical solutions
derived below. Here we make no particular claim for the accuracy or validity of GLEAM as an
evapotranspiration model, nor is our analysis intended to test this. Likewise our analysis should not be
interpreted as implying that GLEAM is any more, or less, susceptible to aggregation bias than other
evapotranspiration schemes, because this question is beyond the scope of the current paper.

**2.2. Mathematical framework for predicting aggregation bias**
*Nonlinear averaging using second-order Taylor expansions*
ET is a nonlinear function of its drivers. An intrinsic property of any nonlinear function is that the average of
the function will not equal the function evaluated at the average inputs (e.g., Rastetter et al., 1992; Giorgi and
Avissar, 1997). Thus averaging over sub-grid heterogeneity in ET drivers, as large-scale land surface models do,
would be expected to lead to biased ET estimates, even if the underlying equations were exactly correct. For
an ET function of three variables, namely $R_n$, $w_w$, and $T$, the mean of the ET function, in terms of the function's
value at the mean of its inputs, can be approximated by the second-order Taylor series expansion of the ET
function (Eq. 6):

$$\overline{ET} \approx \widehat{ET} + \frac{1}{2}\left[\frac{\partial^2 ET}{\partial R_n^2}\text{Var}(R_n) + \frac{\partial^2 ET}{\partial w_w^2}\text{Var}(w_w) + \frac{\partial^2 ET}{\partial T^2}\text{Var}(T)\right]$$

$$+ \frac{\partial^2 ET}{\partial R_n \partial T}\text{Cov}(R_n, T) + \frac{\partial^2 ET}{\partial R_n \partial w_w}\text{Cov}(R_n, w_w) + \frac{\partial^2 ET}{\partial w_w \partial T}\text{Cov}(w_w, T),\qquad(7)$$

where $\overline{ET}$ is the estimate of the true average of the nonlinear ET function over its variable inputs, $\widehat{ET}$ is the ET
function evaluated at its mean inputs, and the derivatives are understood to be evaluated at the mean values



of the variables ($\overline{R_n}, \overline{w_w}, \overline{T}$). For the specific case of the GLEAM model, these derivatives are derived
analytically from the ET function described by Eq. 6, directly yielding the following expressions:

$$\widehat{ET} = \left[-4\overline{w}_w{}^2 + 4.8\overline{w}_w - 0.44\right] * 0.02905 * \frac{a\,e^{b\overline{T}}}{a\,e^{b\overline{T}} + 0.073}\overline{R}_n, \tag{8}$$

$$\frac{\partial^2 ET}{\partial R_n{}^2} = 0, \tag{9}$$

$$\frac{\partial^2 ET}{\partial w_w{}^2} = [-8] * 0.02905 * \frac{\Delta}{\Delta + \gamma}R_n \qquad (w_{wp} \leq w_w \leq w_c), \tag{10a}$$

$$\frac{\partial^2 ET}{\partial w_w{}^2} = 0 \qquad (w_w < w_{wp}, \quad w_w > w_c), \tag{10b}$$

$$\frac{\partial^2 ET}{\partial T^2} = [-4w_w{}^2 + 4.8w_w - 0.44] * 0.02905 * R_n * b^2 * \frac{\gamma^2\Delta - \gamma\Delta^2}{(\gamma + \Delta)^3}, \tag{11}$$

$$\frac{\partial^2 ET}{\partial R_n \partial T} = [-4w_w{}^2 + 4.8w_w - 0.44] * 0.02905 * \frac{\Delta}{\Delta + \gamma} * \frac{b\gamma}{\Delta + \gamma}, \tag{12}$$

$$\frac{\partial^2 ET}{\partial R_n \partial w_w} = [-8w_w + 4.8] * 0.02905 * \frac{\Delta}{\Delta + \gamma} \qquad (w_{wp} \leq w_w \leq w_c), \tag{13a}$$

$$\frac{\partial^2 ET}{\partial R_n \partial w_w} = 0 \qquad (w_w < w_{wp}, \quad w_w > w_c), \tag{13b}$$

$$\frac{\partial^2 ET}{\partial w_w \partial T} = [-8w_w + 4.8] * 0.02905 * \frac{\Delta}{\Delta + \gamma} * \frac{b\gamma}{\Delta + \gamma} * R_n \qquad (w_{wp} \leq w_w \leq w_c), \text{ and} \tag{14a}$$

$$\frac{\partial^2 ET}{\partial w_w \partial T} = 0 \qquad (w_w < w_{wp}, \quad w_w > w_c), \tag{14b}$$

where $\Delta$ depends on temperature as described in Eq. (4). The difference between the average of the functions
($\overline{ET}$) and the function of the averages ($\widehat{ET}$), or, equivalently, the sum of all the other terms in Eq. (7),
represents the aggregation bias. The magnitude of this bias can be calculated by combining Eqs. 7-14 with
estimates of the variances and covariances of the input variables.

The approach outlined in Eq. (7) is general and could be extended to other land surface modeling schemes.
The partial derivatives in Eqs. (8-14), of course, are specific to the GLEAM equations; for other models they
would differ. More complex land surface model algorithms may not have such simple analytical derivatives; in
that case, the derivatives can be evaluated numerically.

**2.3. Sub-grid heterogeneity and aggregation bias in ET estimates across Switzerland**
Drivers of ET (i.e., soil moisture, net radiation, and temperature) can be highly heterogeneous within the grid
cells of typical ESMs. Soil moisture can show pronounced spatial variability, especially in areas where surface
roughness, porosity, and permeability vary by orders of magnitude across a variety of length scales (Giorgi and
Avissar, 1997). Temperature and incoming radiation vary significantly with season, elevation, altitude, and
albedo. Switzerland, for example, shows strong local variations in average annual temperature, soil moisture
content, net radiation, and albedo (Fig. 1; albedo values in Fig. S1).


We quantified how averaging over spatial (and temporal) heterogeneities of ET drivers affects estimated ET at
several grid scales across Switzerland, as an example case for which high-resolution data are available. Our
analysis is based on 500-m input data of temperature (interpolation of MeteoSwiss data after Viviroli et al.,
2009), net radiation (Viviroli et al., 2009), and soil moisture (simulations from the hydrological model PREVAH,
Brunner et al., 2019; Speich et al., 2015; Orth et al., 2015; Zappa et al., 2003) at daily time steps for the 2004
growing season. Although our soil moisture data are derived from model simulations whose accuracy is
difficult to assess due to the scarcity of real-world soil moisture measurements, for our purposes all that is
necessary is that the simulated values exhibit realistically complex spatial variability.

We used the GLEAM equations, as outlined in Sect. 2, to calculate ET for each day at the 500-m resolution of
these input data. We use these 500-m ET estimates as virtual "truth" for the purpose of our analysis, because
our goal is not to determine whether GLEAM estimates of ET are accurate (compared to direct measurements,
for example), but rather to quantify how spatial aggregation affects them.

To quantify how spatial aggregation affects model estimates of ET, we calculated ET over larger spatial scales
in two different ways. First, we averaged the 500-m ET estimates over 1/32, 1/16, 1/8, 0.25, 0.5, 0.75, 1, and 2-
degree grid cells across Switzerland, to represent the "true" average ET at those grid scales. Second, we
averaged the 500-m input data (of temperature, soil moisture, and net radiation) over the same grid cells, and
then used these grid-cell-averaged input data in the GLEAM equations to calculate the modeled coarse-
resolution ET at each grid scale. The deviation of the modeled coarse-resolution ET from the "true" average ET
measures the aggregation bias. Because this numerical experiment uses the same model equations, based on
the same underlying data, for the ET calculations at each spatial resolution, it isolates spatial aggregation as
the only possible cause of the difference between the "true" average ET ($\overline{ET}$ in Eq. 7) and the coarse-resolution
modeled ET ($\widehat{ET}$ in Eq. 7) at each grid scale.

Figure 2a shows that the ET aggregation bias varies considerably across Switzerland, and also varies
considerably with grid scale. The average aggregation bias is higher at coarser grid scales, averaging 21.4% at
2-degree grid resolution and 16.8% at 1-degree grid resolution across all of Switzerland (calculated as the
median of the daily aggregation biases over the growing season; Fig. 2a). Smaller grid scales typically exhibit
smaller aggregation biases (averaging 7% at 1/16-degree grid resolution across all of Switzerland calculated as
the median of the daily aggregation biases over the growing season) because they typically average over less
spatial heterogeneity, but even at the smallest grid scales, aggregation biases can locally exceed 68%, as
indicated by the red grid cells in Fig. 2a. These figures are medians for the entire growing season of 2004; the
aggregation biases of two randomly selected days (May 31[st] and July 21[st], 2004) at several spatial scales are
much larger overestimation of ET in parts of southern Switzerland (Figs. S2, S3).

Using our 500-m input data, we can test how well Eq. (7) estimates the difference between the "true" average
ET and the coarse-resolution modeled ET at each grid scale. We used Eqs. (8-14) to calculate the partial
derivatives of the GLEAM equations for each grid cell and time step, using the grid-cell averaged values of the
input data. We then multiplied these derivatives by the corresponding variances and covariances among the
500-m input data to obtain bias estimates via Eq. (15) for each grid cell and time step:
$$\text{Bias} = \widehat{\text{ET}} - \overline{\text{ET}} \approx -\frac{1}{2}\left[\frac{\partial^2\text{ET}}{\partial R_n{}^2}\text{Var}(R_n) + \frac{\partial^2\text{ET}}{\partial w_w{}^2}\text{Var}(w_w) + \frac{\partial^2\text{ET}}{\partial T^2}\text{Var}(T)\right]$$
$$-\frac{\partial^2\text{ET}}{\partial R_n\partial T}\text{Cov}(R_n,T) - \frac{\partial^2\text{ET}}{\partial R_n\partial w_w}\text{Cov}(R_n,w_w) - \frac{\partial^2\text{ET}}{\partial w_w\partial T}\text{Cov}(w_w,T), \qquad (15)$$

where $\overline{\text{ET}}$ is the true average ET at some grid resolution, $\widehat{\text{ET}}$ is the modeled coarse-resolution ET at the same
spatial scale, the right-hand side is the Taylor expansion estimate of the aggregation bias. We then compared
these estimated biases against the "true" aggregation biases (the difference between the "true" average ET
and the coarse-resolution modeled ET) in the numerical experiment described above. The true bias, in other
words, is $\widehat{\text{ET}} - \overline{\text{ET}}$ in Eq. (15), and the estimated bias is the Taylor approximation on the right-hand side.

Figure 2b shows that the aggregation bias estimated by Eq. (15) is generally similar, in both overall magnitude
and spatial distribution, to the "true" aggregation biases calculated by the numerical experiment. This
comparison is shown more explicitly in Fig. 3, in which the estimated aggregation bias is compared with the
"true" aggregation bias for each grid cell at each grid scale. Figures 2 and 3 show that Eq. (15) is generally a
good predictor of aggregation bias. Both the estimated aggregation biases (Fig. 2) and the "true" aggregation
biases are markedly higher in regions of greater topographic complexity (Fig. S4).

**2.4. Correcting for aggregation bias**
2.4.1.  Identifying drivers of aggregation bias
The Taylor expansion in Eq. (15) not only allows one to quantify the aggregation bias; it also allows one to
quantify the relative importance of the three input variables (net radiation, soil moisture, and temperature) as
drivers of that bias. Each of the terms in Eq. (15) combines a variance or covariance that expresses how
variable the input data are, and a second derivative that expresses how sensitive the average ET is to that
variability. Each of these terms – a derivative multiplied by a variance or covariance – has the same units as ET,
and thus they can be directly compared to one another.

Table 1 shows each of the aggregation bias terms, calculated over all of Switzerland for the two randomly
chosen days mentioned in Sect. 4 (May 31$^{st}$ and July 21$^{st}$, 2004). For these two example days, the aggregation
bias is clearly dominated by a single term, associated with the variance of soil moisture. The variance in net
radiation ($Rn$) creates no aggregation bias, because GLEAM ET is a linear function of Rn; thus positive and
negative deviations from average $Rn$ will increase and decrease ET by exactly offsetting amounts. Similarly, the
variance in temperature ($T$) also results in little aggregation bias, because GLEAM ET increases nearly linearly
with $T$ across a wide range of temperature. The covariance terms similarly lead to little aggregation bias. By
contrast, the strong curvature in the quadratic dependence of ET on soil moisture (Eq. 6) implies that positive



and negative deviations from mean soil moisture will not have offsetting ET effects, and thus that spatial
heterogeneity in soil moisture can significantly alter average ET.

2.4.2.  Correcting for aggregation bias using sub-grid closure relationships
The Taylor expansion framework in Eq. (7) can be used not only to diagnose aggregation bias, but also to
estimate sub-grid closure relationships that correct for the effects of small-scale heterogeneity. The variance
and covariance terms in Eq. (7) express how sub-grid heterogeneity affects average ET at the grid scale,
implying that these aggregation bias estimates could be used to improve grid-scale ET estimates, without
explicitly modeling ET at high resolutions. This approach could be particularly useful in land surface algorithms
that are part of coarser-resolution Earth system models; in such cases it may be much more efficient to
evaluate Eqs. 7-14 at the coarse grid resolution than to directly evaluate the underlying ET model, Eq. 6, at
high resolution. The Taylor expansion approach could also be attractive where we lack spatially explicit high-
resolution maps of the ET drivers, but where their variances and covariances can nonetheless be estimated
from other sources (such as from the variability of topography, mapped soil units, remote sensing data, etc.).

It is beyond our scope here to construct such variance and covariance estimates, but we can illustrate how
they could potentially be used. The solid red symbols in Fig. 4 show the relationships between "true" average
ET and modeled grid-cell-averaged ET, for each grid cell (and one example day, July 21$^{st}$, 2004) at several
different grid scales. For comparison, the open grey symbols in Fig. 4 show average ET estimated by the Taylor
expansion approach of Eq. (7), which corrects for sub-grid heterogeneity effects using only grid-cell-averaged
estimates of the ET drivers and their small-scale variances and covariances.

The heterogeneity-corrected ET estimates shown by the open symbols in Fig. 4 cluster much closer to the 1:1
line than the modeled grid-cell-averaged ET values shown by the solid red symbols, suggesting that the Taylor
expansion approach may substantially improve estimates of grid-cell-averaged ET. Real-world results may be
less clear than those shown in Fig. 4, because the heterogeneity-corrected ET estimates (the open symbols in
Fig. 4) are calculated using exact values for the variances and covariances of the ET drivers within each grid
cell, and in real-world cases these variances and covariances will not be known precisely. Figure 4 nonetheless
demonstrates the potential value of knowing, or being able to estimate, those variances and covariances.
Efforts to determine those variances and covariances can be focused on the terms that matter the most, if one
can identify the main drivers of aggregation bias using the methods described in Sect. 5.1 above.



Table 1. Relative importance of different ET drivers in aggregation bias estimates (different terms in Eq. 15). Va
lues are calculated for all of Switzerland for the two randomly chosen days (May 31st and July 21st, 2004). The
aggregation bias is dominated by the term associated with the variance of soil moisture for these two example
days.

| | $\widehat{ET}$ mm d$^{-1}$ | $\overline{ET}$ mm d$^{-1}$ | Bias % | Contribution of $Var(R_n)$ term in % aggregation bias (%) | Contribution of $Var(w_w)$ term in % aggregation bias (%) | Contribution of $Var(T)$ term in % aggregation bias (%) | Contribution of $Cov(R_n,T)$ term in % aggregation bias (%) | Contribution of $Cov(R_n,w_w)$ term in % aggregation bias (%) | Contribution of $Cov(R_n,w_w)$ term in % aggregation bias (%) |
|---|---|---|---|---|---|---|---|---|---|
| Calculation | (Eq. 8) | (Eq. 7) | (Eq. 15) | $\dfrac{\frac{1}{2}\frac{\partial^2 ET}{\partial R_n^2}Var(R_n)}{(\widehat{ET}.Bias)}$ | $\dfrac{\frac{1}{2}\frac{\partial^2 ET}{\partial w_w^2}Var(w_w)}{(\widehat{ET}.Bias)}$ | $\dfrac{\frac{1}{2}\frac{\partial^2 ET}{\partial T^2}Var(T)}{(\widehat{ET}.Bias)}$ | $\dfrac{\frac{\partial^2 ET}{\partial R_n\partial T}Cov(R_n,T)}{(\widehat{ET}.Bias)}$ | $\dfrac{\frac{\partial^2 ET}{\partial R_n\partial w_w}Cov(R_n,w_w)}{(\widehat{ET}.Bias)}$ | $\dfrac{\frac{\partial^2 ET}{\partial w_w\partial T}Cov(w_w,T)}{(\widehat{ET}.Bias)}$ |
| 31.05.2004 | 2.658 | 1.636 | 38.43 | 0 | 97.94 | 0.04 | 0.48 | 0.22 | 1.32 |
| 21.07.2004 | 2.38 | 1.685 | 29.16 | 0 | 94.67 | 1.05 | 3.03 | 0.33 | 0.92 |


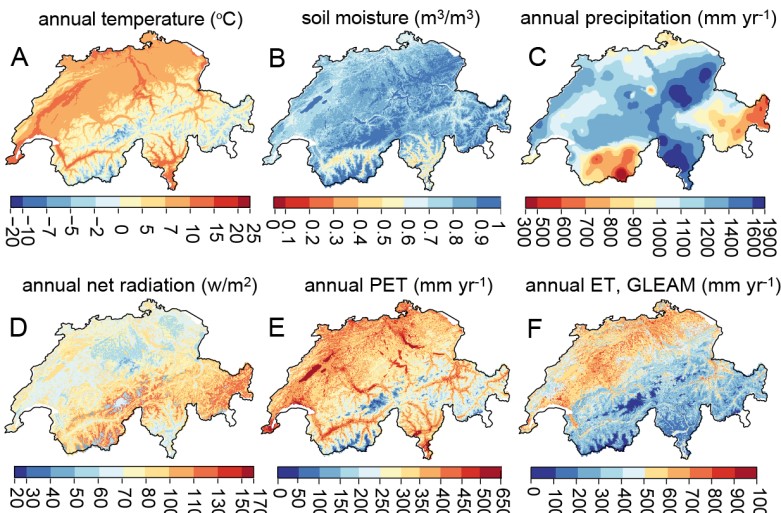

annual temperature (°C)    soil moisture (m³/m³)    annual precipitation (mm yr⁻¹)
annual net radiation (w/m²)    annual PET (mm yr⁻¹)    annual ET, GLEAM (mm yr⁻¹)


Figure 1. Spatial distribution of input data for the year 2004 at 500-m resolution: Annual mean (A) temperature
(°C), (B) soil moisture (m³ m⁻³, simulated by the PREVAH hydrological model), (C) precipitation (mm yr⁻¹), (D)
net radiation (W m⁻²), (E) potential evapotranspiration (PET, mm yr⁻¹) using the Priestley-Taylor equation (Eq.



3), and (F) evapotranspiration (ET, mm yr$^{-1}$) using the approach used in the GLEAM model (Eq. 1). See Table. S1
for references.


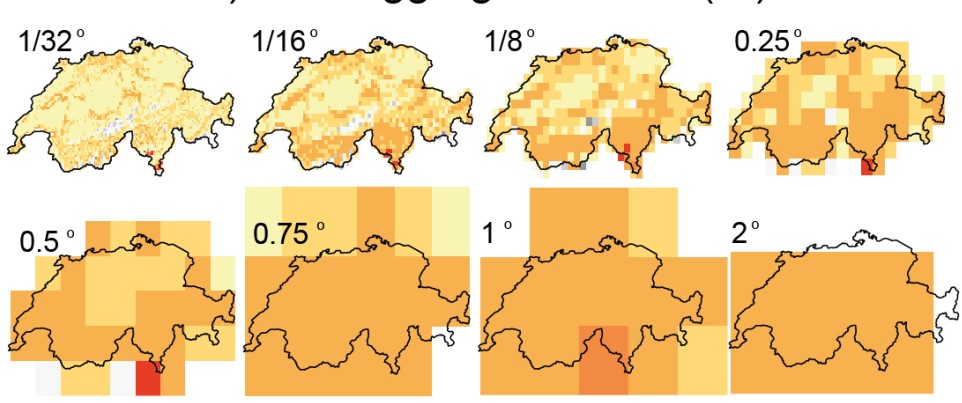

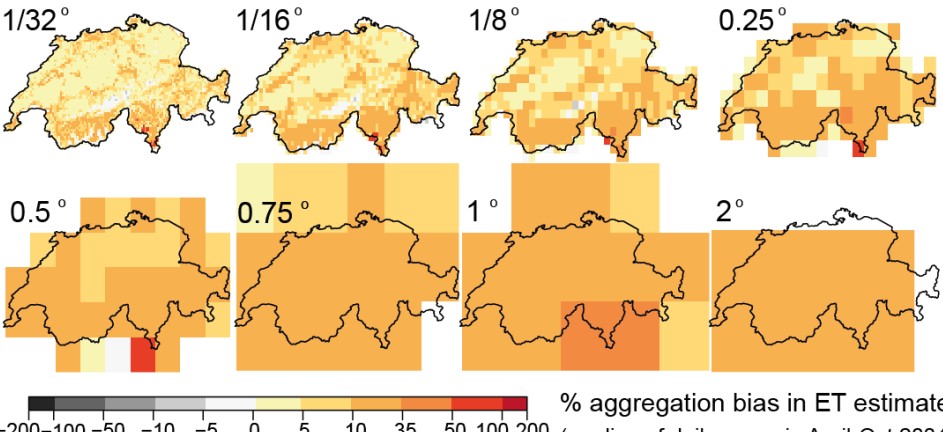


Figure 2. a) "True" aggregation bias in ET, as calculated by averaging the 500-m resolution ET estimates using
fine-resolution input data in Eq. 6, over 1/32, 1/16, 1/8, 0.25, 0.5, 0.75, 1, and 2-degree grid cells across
Switzerland. b) Aggregation bias in ET, as estimated by Eq. 7 from grid-cell averaged temperature (°C), soil
moisture ($w_w$), net radiation ($R_n$), their variances at each grid scale, and the covariances of all pairs of variables
using the 500-m input data. At finer grid scales, the aggregation bias is more localized, and smaller on average.
Across Switzerland as a whole, average aggregation bias becomes smaller as grid scales become finer, but
never disappears completely.

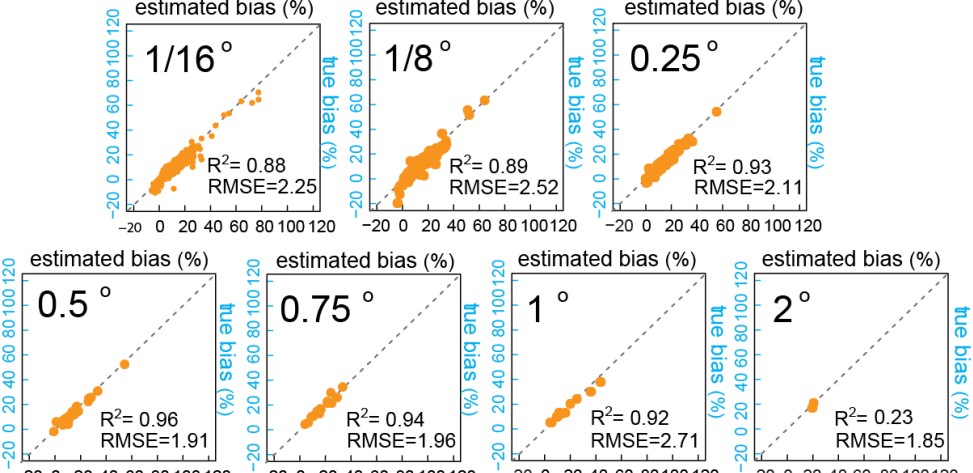


Figure 3. Daily estimated aggregation bias in ET estimates (%, median of daily biases in Apr.-Oct. 2004) versus

daily true aggregation bias in ET estimates (%, median of daily biases in Apr.-Oct. 2004) at several spatial

scales. Estimated aggregation biases are calculated using Eq. 7. True aggregation biases are calculated as

differences between the finer resolution ET estimates from finer resolution input data, averaged over several

spatial scales (average of functions) and ET values calculated from average inputs at each spatial scale

(function of averages). The coefficients of determination ($R^2$) between the true and estimated aggregation

biases verify the reliability of the Taylor expansion method and Eq. 7 as estimates of the aggregation bias.

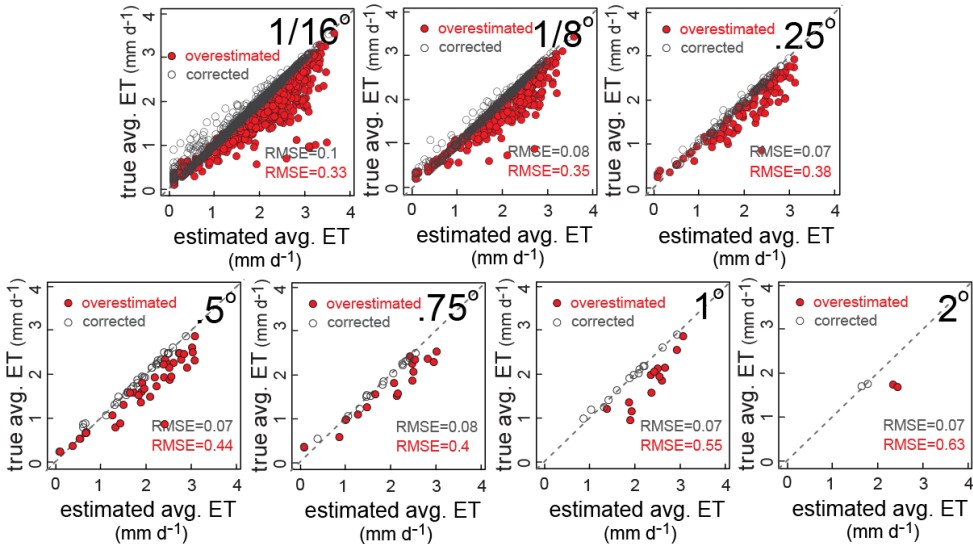


Figure 4. Daily estimated ET rates versus "true" average ET at each grid cell at several different grid scales
(example day, July 21st, 2004). The solid red symbols demonstrate the relationships between "true" average ET
calculated using fine-resolution data at each grid cell and modeled grid-cell-averaged ET using grid-cell-



averaged inputs in Eq.8, for each grid cell at several different grid scales (overestimated). For comparison, the
open symbols show true average versus average ET estimated by the Taylor expansion approach of Eq. (7),
which corrects for sub-grid heterogeneity effects using only grid-cell-averaged estimates of the ET drivers and
their small-scale variances and covariances (heterogeneity-corrected ET estimates, corrected).

**3. Discussion**
Averaging over spatially heterogeneous ET drivers leads to substantial aggregation biases in ET flux estimates
from a typical mechanistic large-scale land surface model. This aggregation bias arises from the inherent
nonlinearities in evapotranspiration processes, coupled with the inherent spatial heterogeneity in the driving
factors. The joint effects of these nonlinearities and heterogeneities can be estimated using second-order
Taylor expansions of the governing equations. Using Switzerland as a test case, we have shown that median
aggregation biases of 10-35% are common, even at grid scales substantially smaller than those typically used in
land surface models (Fig. 2). These biases can be much larger for individual days (Figs. S2 and S3). These biases
can have substantial consequences for water and energy flux estimates in land surface models and
consequently for temperature predictions in coupled models. The overestimated evaporative fluxes would
lead to overestimated latent heat fluxes and underestimated sensible heat fluxes, and thus potentially to
underestimates of expected temperature increases in a changing climate. Unrealistically high evaporation
estimates lead to cooler modeled temperatures and wetter modeled climates. Correcting for the aggregation
bias in ET fluxes would lead to reduced evaporative cooling and increased atmospheric heating via sensible
heat flux.

In coupled Earth system models, ET fluxes influence how surface temperature, net radiation, and soil moisture
evolve through time, and thus influence future values of ET. The analyses shown in Figs. 2-4 are based on static
values for each day, and thus do not account for the propagation of aggregation biases forward through time.
Estimating the consequences of aggregation biases for dynamic modeling would require fully coupled Earth
system model simulations rather than the single ET algorithm analyzed here. In a dynamic model, the Taylor
expansion approach can potentially be used to correct for aggregation biases in each time step, using
statistical models for the variances and covariances of the ET drivers. Thus, estimating aggregation biases in a
dynamic model would not require explicitly simulating sub-grid heterogeneity at every time step. Correcting
for aggregation biases at each modeling time step would prevent them from propagating further into future
time steps, or into the partitioning of future water and energy fluxes at the land surface. The present paper
does not illustrate this dynamic correction for aggregation biases, but establishes the theoretical framework
for it.

The purpose of our analysis was to demonstrate how aggregation bias due to spatial heterogeneity can be
quantified (Sects. 3-4), how its dominant drivers can be identified (Sect. 5.1), and how its effects can be





efficiently corrected for, using sub-grid closure relationships (Sect. 5.2). For this demonstration, we chose GLEAM as an illustrative example, and Switzerland as a topographically complex case study where high-resolution data on the ET drivers are available. Applications of this approach to more complex land surface models may require calculating the necessary derivatives (see Eq. 7) numerically rather than analytically, and applications where high-resolution data are unavailable may require statistically estimating the variances and covariances among the drivers of ET, based on their relationships with topography, soil types, land cover, etc. Using the approach outlined here, one can account for the effects of sub-grid heterogeneity without explicitly modeling ET at fine spatial resolution, which could be impractical due to computational costs, or impossible due to a lack of fine-resolution input data.

In our analysis, spatial heterogeneity in soil moisture emerged as the dominant driver of aggregation bias in ET estimates. Particularly if this result can also be confirmed in other regions and climates, it points to the importance of improving our understanding of spatial patterns of soil moisture and what controls them. The lower topographic curvature of coarsely gridded landscapes can lead models to predict higher soil moisture at coarser grid scales (Kuo et al., 1999); higher soil moisture at larger grid scales would lead to even higher modeled values of ET, beyond the effects of the aggregation biases analyzed here. Soil moisture may also be substantially influenced by lateral subsurface transfers of water, which are ignored in our analysis and are also ignored by many land surface models. Overlooking lateral transfers could potentially bias ET estimates in large-scale land surface models (Fan et al., 2019), but this is beyond the scope of the present study.

**Acknowledgements**

We thank Prof. Ying Fan Reinfelder for numerous insightful discussions and for helpful comments on the manuscript. E.R.F. acknowledges support from the Swiss National Science Foundation (SNSF) under Grant No. P2EZP2_162279.

**Data Availability Statement**

We will upload the source data for this study to a FAIR repository and provide the URL with the final version of the paper.



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
