# Peer review of "Averaging over spatiotemporal heterogeneity substantially biases evapotranspiration rates in a mechanistic"

_Hydrology and Earth System Sciences, 2020_

## Referee Comment (RC1) · Anonymous Referee #1 · 12 Mar 2020

In their study, the authors adapt a general mathematical method that was published by them earlier (2017) that can be used to determine and correct the biases related to the spatial aggregation of modeled, gridded evapotranspiration fields. The method is exemplarily applied for Switzerland, based on the GLEAM evapotranspiration model. I consider the contribution as innovative and as relevant for the field of hydrometeorological modeling and I recommend its publication after the following points were adequately addressed:

General comments

[Figure]

Is it always that with higher resolution data models give more realistic estimates of ET? In the introduction you mainly address biases caused by rescaling of ET fields, but how does that rely to observations? Is there evidence in literature for the assumption that higher resolution data usually provides more realistic rates?

You use GLEAM to prove your concept. But looking at the comparisons of true and estimated biases in Fig. S2 and S3, it seems that your approach does not work well for resolutions smaller than 0.25° (which is the target resolution of GLEAM). So maybe GLEAM is kind of optimized to this resolution and is not too realistic for higher ones? How would you explain the increased scatter between true and estimated biases for the 1/32° and 1/16° resolutions?

Specific comments:

16: I would say that the drivers for droughts and heatwaves are precipitation, radiation, wind, temperature and soil moisture but not ET. Heatwaves occur because of the advection of warm and dry air. Droughts are caused by lacking precipitation.

42: Can you give a rough number (in percent) of typical deviations?

140: Priestley-Taylor was already cited before in L 101.

167-173: You should cite your 2017 paper here again, is is cited in the introduction but when I read the equations below a quick link to where they have been derived would be helping; also you should explain shortly the meaning of the variance and covariance terms here. They are only explained in L 246.

177-179: Eq. 8 is not a derivative

179-188: Why was the interception term of Eq. 6 been skipped in the derivative calculations?

221-230: What algorithm was used for averaging?

271-280: Are there dates where other variables than soil moisture have an increased

impact?

309, 390, 391: The section references seem to be broken.

Fig. S2a) / S3a), please put the 6 maps into two rows, the color key numbers are hard to read

References: unify format, many DOIs are missing, some are printed as links, some have no preceding "DOI" (pleas stick to HESS typesetting rules); Use en-dash for page ranges instead of simple dash 522: "Uber" -> "Über"

Minor:

15: feedbacks -> feedback 124: please change to "I is interception loss" or "I are interception losses" 367: two times "These biases can" maybe replace by "and"
* * *

---

## Referee Comment (RC2) · Anonymous Referee #2 · 25 Mar 2020

In this paper, the authors quantified and corrected the aggregation bias resulting from spatial heterogeneity in evapotranspiration (ET) estimates in a land evaporation model using the second-order Taylor expansions mathematical framework, an approach published by the authors previously in 2017. The GLEAM land surface model was chosen as its governing equations for calculating ET (Priestley-Taylor method) were amenable to analytical instead of numerical solutions and Switzerland was selected as the study area where high-resolution data (500m) on the ET drivers are available. This work is interesting and has important implications for Earth System Models. It can be accepted

after several comments are addressed.

General comments

In Figures 3 and 4, the graph for 1/32 degree seems missing. Moreover, Figures S2 and S3 (two selected days) indicate that the result shown in graph (1/32 degree) is not as good as other coarser resolutions, what is the possible reason for this?

The soil moisture plotted in Figure 1(B), S2(a) and S3(a) stands for the volumetric soil moisture (should be smaller than soil porosity) or soil moisture saturation (i.e. volumetric soil moisture/soil porosity, ranging from 0 and 1)? In addition, because spatial heterogeneity in soil moisture is found as the dominant driver of aggregation bias in ET estimates, perhaps the authors can provide the corresponding spatial distribution graph of soil moisture across different grid scales by averaging the 500m soil moisture in the supporting information.

Specific comments

Lines 58-61, it will be much clearer to the readers if the authors cite separately which literature found 'increases in average ET' and which literature reported 'decreases in grid-cell average ET'.

Line 117, 0.25-degree spatial resolution (i.e. corresponding to what kilometers?).

Line 156 and Line 174, compared equation (6) and (7), the interception term (containing information about precipitation) is gone, why? Especially considering that this interception term is important as shown in Figure 1(E) and 1(F) as well as Figures S2(a) and S3(a).

Lines 222-224, how did the authors conduct the "average" algorithm?

Table 1, the two example days showed that variance of soil moisture is the dominant driver of aggregation bias in ET estimates, is this true for all the other days?

Technical corrections

[Figure]

Lines 309, 390, 381, section 5.1 and 5.2 is typo.

Please also note the supplement to this comment:
https://www.hydrol-earth-syst-sci-discuss.net/hess-2020-46/hess-2020-46-RC2-supplement.pdf

---

## Author Response (AR1)

**Hydrol. Earth Syst. Sci.**
**hess-2020-46**

**Dear Reviewer #1,**
**Thank you for your review and the detailed comments. Below please find our point by point response to your suggestions and questions. The Reviewer's comments are in regular font and our response is in bold.** The line numbers correspond to the revised manuscript in "all mark up" view setting.

**Response to Referee #1**
In their study, the authors adapt a general mathematical method that was published by them earlier (2017) that can be used to determine and correct the biases related to the spatial aggregation of modeled, gridded evapotranspiration fields. The method is exemplarily applied for Switzerland, based on the GLEAM evapotranspiration model. I consider the contribution as innovative and as relevant for the field of hydrometeorological modeling and I recommend its publication after the following points were adequately addressed:

**We thank the reviewer for his/her interest in this work.**

General comments
Is it always that with higher resolution data models give more realistic estimates of ET? In the introduction you mainly address biases caused by rescaling of ET fields, but how does that rely to observations? Is there evidence in literature for the assumption that higher resolution data usually provides more realistic rates? You use GLEAM to prove your concept. But looking at the comparisons of true and estimated biases in Fig. S2 and S3, it seems that your approach does not work well for resolutions smaller than 0.25 (which is the target resolution of GLEAM). So maybe GLEAM is kind of optimized to this resolution and is not too realistic for higher ones? How would you explain the increased scatter between true and estimated biases for the 1/32 and 1/16 resolutions?

> **First of all, it is important to remember that we are not comparing GLEAM with real-world measurements and therefore we cannot evaluate the realism of GLEAM at any resolution. We are not assuming that higher-resolution data is more realistic; instead, we use the higher-resolution estimates as a benchmark for synthetic experiments that examine how these ET estimates change with aggregation scale. As we note on lines 217-219, we use 500-m ET estimates (derived from GLEAM) as virtual "truth" and then see how these estimates, averaged over a range of larger scales, compare with GLEAM estimates of ET obtained from averages of temperature, net radiation, and soil moisture over those same larger scales.**

> **We further looked into the point raised by the reviewer regarding the increased scatter between true and estimated biases for the 1/32 and 1/16 resolutions plots of figures S1 and S2. We noticed that due to a coding error, equations 10b, 13b, and 14b were not implemented correctly, meaning that the stress factor function was considered nonlinear in the full range of soil moisture and not only when soil moisture was between 0.1 and 0.6.**

> **The stress factor function is nonlinear between volumetric soil moisture values of 0.1 and 0.6 as it is defined in GLEAM, and is equal to 0 or 1 outside this soil moisture range. Therefore the first and second derivatives of the ET function with regard to soil moisture are equal to 0 outside this range (eq10b, 13b, and 14b). Unfortunately we noticed that this point was overlooked in our original calculations in the code and the stress factor function was mistakenly considered as a nonlinear function for the entire range of soil moisture. We have**

now corrected this glitch and verified that the script is handling the 0.1 and 0.6 soil moisture conditions and the corresponding variability of soil moisture in this range correctly. The supplementary figures corresponding to estimated averaging error versus true averaging error for the two days also exhibit much less scatter than before. In fact, with this correction the $R^2$ of the scatter plot of the 1/32 degree resolution increases to 0.91 on May 31$^{st}$ 2004 and 0.97 on July 21$^{st}$ 2004 after this correction. We will rerun the script and redraw all the figures in the revised manuscript.

After correcting this glitch, the estimated aggregation biases in Figures S1 and S2 were quite close to the one-to-one line for almost all the points, regardless of the resolution. This indicates that our method for predicting the aggregation bias generally works well. At the highest resolutions (smallest grid cells), however, there are a few cells that lie farther from the 1:1 line. These correspond to individual points in which the absolute values of ET are very small (snow-covered or glacierized landscapes), so even small prediction errors can appear as large percentage errors. But because these large percentage prediction errors are small in absolute terms, they mostly disappear when they are aggregated to larger grid cells. Thus the mean averaging error across Switzerland decreases sharply (almost exponentially) as the resolution increases.

Figure 2, 3, 4, S1, S2 and the related statements in the text have been changed throughout the manuscript.

Specific comments:
16: I would say that the drivers for droughts and heatwaves are precipitation, radiation, wind, temperature and soil moisture but not ET. Heatwaves occur because of the advection of warm and dry air. Droughts are caused by lacking precipitation. 42: Can you give a rough number (in percent) of typical deviations?

We will correct this statement to: "Due to its feedbacks to large-scale hydrological processes and its impact on atmospheric dynamics, ET is one of the drivers of droughts and heatwaves".

Line 16 has changed to "ET is one of the drivers of droughts and heatwaves".

With regard to the typical deviations, paragraph 2 and 3 of the introduction (Lines 47-70) report the direction and magnitude of these deviations according to a few studies.

140: Priestley-Taylor was already cited before in L 101.
This citation is directly relevant to the PET formula and we found it is helpful to keep it where the equation is presented.

167-173: You should cite your 2017 paper here again, is cited in the introduction but when I read the equations below a quick link to where they have been derived would be helping; also you should explain shortly the meaning of the variance and covariance terms here. They are only explained in L 246.

We added the "Rouholahnejad Freund and Kirchner, 2017" paper as a reference (Line 172) and edited the explanation for equation 7 as: where $\overline{ET}$ is the estimate of the true average of the nonlinear ET function over its variable inputs, $\widehat{ET}$ is the ET function evaluated at its mean inputs, and the derivatives are understood to be evaluated at the mean values of the variables ( $\overline{R_n}, \overline{w_w}, \overline{T}$ ) and multiplied by the corresponding variances and covariances among the finer-resolution input data." (Lines 178-181)

177-179: Eq. 8 is not a derivative

We corrected the corresponding statement to make this point clearer: "For the specific case of the GLEAM model, the ET function is evaluated at its mean inputs ((ET)ˇ) and these derivatives are derived analytically from the ET function described by Eq. 6, directly yielding the following expressions:" (Lines 181-183)

179-188: Why was the interception term of Eq. 6 been skipped in the derivative calculations?

In GLEAM, interception loss is explicitly modelled according to Gash's analytical model (Gash, 1979; Valente et al., 1997). Following this approach, the volume of water that evaporates from the canopy is estimated as a linear function of the daily rainfall using parameters that describe the canopy cover, canopy storage, and mean rainfall and evaporation rate during saturated canopy conditions.

Because the interception loss in GLEAM is a linear function of amount of rainfall necessary to saturate the canopy, it has negligible effects on the aggregation bias.

We added a statement to explain this point (Lines 197-199):
"Note that the interception term in equation 6 is dropped out from the derivatives as the interception loss in GLEAM is a linear function of amount of rainfall necessary to saturate the canopy and therefore has negligible effect when averaged."

221-230: What algorithm was used for averaging?

-These are pure arithmetic averages (sum of values divided by number of values).

271-280: Are there dates where other variables than soil moisture have an increased impact?

The terms have different positive and negative contributions (increasing or decreasing effects on total bias) on the two days, with some of the variance and covariance terms being negative or positive. For example, the Rn and SM covariance term on May 31st 2004 is slightly negative (-0.53) but this same term is slightly positive on July 21st 2004 (0.88).

On most of the days of the year 2004, the soil moisture variance term is the dominant driver of the aggregation bias. However, there are some days in which other factors such as the T and Rn covariance term is the dominant factor (e.g, on days 285 and 297 of the year 2004, the T and Rn covariance term constitutes 74.5 % and 90.2 % of the aggregation bais).

309, 390, 391: The section references seem to be broken.
Thanks for pointing this out. We will revise the section numbers.

Fig. S2a) / S3a), please put the 6 maps into two rows, the color key numbers are hard to read

**We will do that.**

We redraw Figures S2a and S3a as requested (supplementary material, Figures S2 and S3)

References: unify format, many DOIs are missing, some are printed as links, some have no preceding "DOI" (pleas stick to HESS typesetting rules); Use en-dash for page ranges instead of simple dash

**We will do that.**

Line 468-561 are revised.

522: "Uber" -> "Über"

**We will revise "Uber" to "Über"**

"Uber" revised to "Über" (Line 552)

Minor:

15: feedbacks -> feedback 124: please change to "I is interception loss" or "I are interception losses" 367: two times "These biases can" maybe replace by "and"

**We will revise these points.**

Line 127 "interception losses" revised to "interception loss"

Lines 378-380 are revised to:

**"These biases can** be much larger for individual days (Figs. S2 and S3) **and potentially** have substantial consequences for water and energy flux estimates in land surface models and consequently for temperature predictions in coupled models."

Hydrol. Earth Syst. Sci.
hess-2020-46

**Dear Reviewer #2,**
**Thank you for your review and the detailed comments. Following please find our point by point response to your suggestions and questions. The Reviewer's comments are in regular font and our response is in bold.**
**The line numbers correspond to the revised manuscript in "all mark up" view setting.**

**Response to Referee #2**
In this paper, the authors quantified and corrected the aggregation bias resulting from spatial heterogeneity in evapotranspiration (ET) estimates in a land evaporation model using the second-order Taylor expansions mathematical framework, an approach published by the authors previously in 2017. The GLEAM land surface model was chosen as its governing equations for calculating ET (Priestley-Taylor method) were amenable to analytical instead of numerical solutions and Switzerland was selected as the study area where high-resolution data (500m) on the ET drivers are available. This work is interesting and has important implications for Earth System Models. It can be accepted after several comments are addressed.

**We thank the reviewer for his/her interest in this work.**

General comments
In Figures 3 and 4, the graph for 1/32 degree seems missing. Moreover, Figures S2 and S3 (two selected days) indicate that the result shown in graph (1/32 degree) is not as good as other coarser resolutions, what is the possible reason for this?

**We looked into the point raised by the reviewer regarding the increased scatter between true and estimated biases for the 1/32 resolution plots of figures S1 and S2. We noticed that due to a coding error, equations 10b, 13b, and 14b were not implemented correctly, meaning that the stress factor function was considered nonlinear in the full range of soil moisture and not only when soil moisture is between 0.1 and 0.6.**

**The stress factor function is nonlinear between volumetric soil moisture values of 0.1 and 0.6 as it is defined in GLEAM, and is equal to 0 or 1 outside this soil moisture range. Therefore the first and second derivatives of ET function with regard to soil moisture are equal to 0 (eq10b, 13b, and 14b). Unfortunately we noticed that this point was overlooked in our original calculations in the code and the stress factor function was mistakenly considered as a nonlinear function for the entire range of soil moisture. We have now corrected this glitch and verified that script is handling the 0.1 and 0.6 soil moisture conditions and the corresponding variability of soil moisture in this range correctly. The supplementary figures corresponding to estimated averaging error versus true averaging error for the two days also exhibit much less scatter than before. In fact, with this correction the $R^2$ of the scatter plot of the 1/32 degree resolution increases to 0.91 on May 31st 2004 and 0.97 on July 21st 2004 after this correction. We will rerun the script and redraw all the figures in the revised manuscript.**

**After correcting for this mistake, the estimated aggregation biases in Figures S1 and S2, were quite close to the one-to-one line for almost all the points, regardless of the resolution. This**

**indicates that our method for predicting the aggregation bias generally works well. At the highest resolutions (smallest grid cells), however, there are a few cells that lie farther from the 1:1 line. These correspond to individual points in which the absolute values of ET are very small (snow-covered or glacierized landscapes), so even small prediction errors can appear as large percentage errors. But because these large percentage prediction errors are small in absolute terms, they mostly disappear when they are aggregated to larger grid cells. Thus the mean averaging error across Switzerland decreases sharply (almost exponentially) as the resolution increases.**

Figure 2, 3, 4, S1, S2 and the related statements in the text have been changed throughout the manuscript.

The soil moisture plotted in Figure 1(B), S2(a) and S3(a) stands for the volumetric soil moisture (should be smaller than soil porosity) or soil moisture saturation (i.e. volumetric soil moisture/soil porosity, ranging from 0 and 1)? In addition, because spatial heterogeneity in soil moisture is found as the dominant driver of aggregation bias in ET estimates, perhaps the authors can provide the corresponding spatial distribution graph of soil moisture across different grid scales by averaging the 500m soil moisture in the supporting information.

**We will add the figure to the supplementary material**
Figure 1(B), S2(a) and S3(a) show volumetric soil moisture ranging from 0 to 1. The spatial distribution of soil moisture averages at several grid scales for the two randomly selected days are now added to the supplementary information (Figures S5 and S6).

Specific comments
Lines 58-61, it will be much clearer to the readers if the authors cite separately which literature found 'increases in average ET' and which literature reported 'decreases in grid-cell average ET'.

**We will cite the literature which reported decreases or increases in average ET separately in the revised manuscript.**
Line 59-63 are revised to:
"Several studies have reported increases in average evapotranspiration (ET) (e.g., Kuo et al., 1999; Boone and Wetzel, 1998; Hong et al., 2009; McCabe and Wood, 2006; El Maayar and Chen, 2006), and at least one has reported decreases in grid-cell average ET (Ershadi et al., 2013), as model grids are coarsened and less spatial heterogeneity is accounted for."

Line 117, 0.25-degree spatial resolution (i.e. corresponding to what kilometers?).

**0.25 degrees is about 27.6 km in the north-south direction and 18.9 km in the east-west direction at the latitude of Switzerland.**

Line 156 and Line 174, compared equation (6) and (7), the interception term (containing information about precipitation) is gone, why? Especially considering that this interception term is important as shown in Figure 1(E) and 1(F) as well as Figures S2(a) and S3(a).

**In GLEAM, interception loss is explicitly modelled according to Gash's analytical model (Gash, 1979; Valente et al., 1997). Following this approach, the volume of water that evaporates from the canopy is estimated as a linear function of the daily rainfall using parameters that describe the canopy cover, canopy storage, and mean rainfall and evaporation rate during saturated canopy conditions.**

**Because the interception loss in GLEAM is a linear function of amount of rainfall necessary to saturate the canopy, it has negligible effects on the aggregation bias.**

We added a statement to explain this point (Lines 197-199): "Note that the interception term in equation 6 is dropped out from the derivatives as the interception loss in GLEAM is a linear function of amount of rainfall necessary to saturate the canopy and therefore has negligible effect when averaged."

Lines 222-224, how did the authors conduct the "average" algorithm?
   **These are pure arithmetic averages (sum of values divided by number of values).**

Table 1, the two example days showed that variance of soil moisture is the dominant
driver of aggregation bias in ET estimates, is this true for all the other days?

   **We re-ran the analysis for the entire Switzerland for every day of the year 2004. In most of the days of the year 2004, soil moisture variance term is the dominant driver of the aggregation bias. However, there are some days in which other factors such as the T and Rn covariance term is the dominant factor (e.g, days 285 and 297 of the year 2004, the T and Rn covariance term constitutes 74.5 % and 90.2 % of the aggregation bias).**

Technical corrections
Lines 309, 390, 381, section 5.1 and 5.2 is typo.

   **OK.**

[revised manuscript text omitted]

---

## Author Response (AR2)

**Hydrol. Earth Syst. Sci.**
**hess-2020-46**

**Dear Editor,**
**Thank you for your comments. Below please find our point by point response to your comments and questions. The Editor's comments are in regular font and our response is in bold. The line numbers correspond to the revised manuscript (rev2) in "all mark up" view setting.**

Editor Decision: Publish subject to minor revisions (review by editor) (25 Aug 2020) by Anke Hildebrandt
Comments to the Author:
Dear authors, thank you for your revisions and new version of the manuscript. I have some few requests that should be quick to accomodate, before the manuscript is ready for publication.

Some of the questions of the reviewers were only answered in the response, but no additional information was added to the manuscript. Could you please add this information?

For example, both reviewers requested which procedure was used for averaging. Could you please enter this information at a suitable place into the manuscript?

> **We added the statements to the manuscript: lines 118-120, 228-230**

Similarly, both reviewers were wondering, whether soil moisture variance only was a strong driver on those days selected for presentation.

> **We now added the statements that was initially only provided to the reviewers to the manuscript: lines 289-291**

Myself, I am wondering about the presentation of the „randomly selected day". How did you select them „randomly"? Can you please add this information to the manuscript.

> **The two selected days are days 150 and 200 of Julian day calendar of year 2004 and were selected arbitrarily. We added this information to the manuscript, changed "randomly" to "arbitrarily" as it conveys better the way we selected the days, and we corrected for the actual date of these two days. (The codes are counting day 150 and 200 of the year 2004, so no miscalculation there). Lines 247-249, and throughout the text and figures.**

Finally, reviewer #2 requested, whether you are showing volumetric soil moisture (the more common term is volumetric soil water content, which ranges from 0 to porosity) or soil moisture saturation (which ranges from 0 to 1). In your response you state that you are presenting soil volumetric soil moisture which ranges from 0 to 1. This must be an error somewhere. From Fig 1 it appears you are presenting soil moisture saturation. Would you please have another look at this?

> **Thank you for picking on this again. Indeed you are right. The data we have used and mapped is soil moisture saturation. We corrected for this in the manuscript (Lines 132-134) and in Figure 1, and S2, S3, S5, and S6.**

I am looking forward to the new version of you manuscript,

**Thank you very much for taking time.**

Anke Hildebrandt

[revised manuscript text omitted]

- Figure S1, Land cover map of Switzerland with corresponding albedo values
- Figures S2-S6
- Data availability

Table S1. *PREVAH* hydrological and meteorological data. All data are in gridded format and at 500 m spatial resolution and (if relevant) daily temporal resolution

| Data | Source |
| --- | --- |
| *PREVAH* soil moisture ($m^3/m^3$) | Simulations from *PREVAH* hydrological model, Brunner et al., 2019 ; Speich et al., 2015; Orth et al., 2015; Zappa et al., 2003 |
| precipitation (mm d-1) | Interpolation of MeteoSwiss data after Viviroli et al., 2009 |
| radiation (W m-2) | Interpolation of MeteoSwiss data after Viviroli et al., 2009 |
| relative humidity (-) | Interpolation of MeteoSwiss data after Viviroli et al., 2009 |
| sun duration (hr) | Interpolation of MeteoSwiss data after Viviroli et al., 2009 |
| temperature (°C) | Interpolation of MeteoSwiss data after Viviroli et al., 2009 |
| vapor pressure (Pa) | Interpolation of MeteoSwiss data after Viviroli et al., 2009 |
| CH land use | BFS, Swiss Federal Statistical Office, 1995 |

[Figure]

Figure S1. Land cover map of Switzerland at 500-meter resolution along with the albedo values associated with each land cover type (BFS, 1995; Viviroli et al., 2009)

**May 31st, 2004**

[Figure]

[Figure]

**May 29th, 2004**

Figure S2. a) Spatial distribution of input data at 500 m resolution for a arbitrarily selected day (29.05.2004) to calculate ET. Potential evapotranspiration (PET, mmyr$^{-1}$) is calculated using the Priestley-

Taylor equation (Eq. 3), and evapotranspiration (ET, mmyr$^{-1}$) is calculated using the approach used in the

GLEAM model (Miralles et al., 2011; Martens et al., 2017; Eq. 1). b) Aggregation bias estimated from 500 m temperature (°C), soil moisture ($w_w$), net radiation ($R_n$), their variances at each grid scale, and the covariances of all the pairs using Eq. 7. Even at the finest resolutions (1/32 and 1/16 degrees) the aggregation bias rises to

50-100 % overestimation in daily ET estimates in South Switzerland. c) Daily approximated aggregation bias in

ET estimates versus daily true aggregation bias in ET estimates at several spatial scales for 29.05.2004.

Approximated aggregation bias is calculated using Eq.7. The true bias is the difference between the finer- resolution ET estimates from finer-resolution input data, averaged over several spatial scales (average of functions) and average ET estimated from average inputs at each spatial scale (function of averages). The coefficient of determination ($R^2$) between the true and approximated aggregation bias confirms the appropriateness of the proposed method and Eq. 7 for approximating the aggregation bias.

[Figure]

**July 21st, 2004**

**a)** temperature (°C)  soil moisture (m³/m³)  precipitation (mmd⁻¹)

global radiation (w/m²)  PET, Priestley & Taylor (mmd⁻¹)  ET GLEAM (mmd⁻¹)

**b)** 1/32°  1/16°  1/8°  0.25°

0.5°  0.75°  1°  2°

% estimated aggregation bias in ET estimates

**c)**

[Figure]

**July 18th, 2004**

a)

temperature (°C)

soil moisture saturation (-)

precipitation (mmd⁻¹)

−20 −10 −7 −5 −2 0 5 7 10 15 20 25

0.1 0.2 0.3 0.4 0.5 0.6 0.7 0.8 0.9 1

1.2 2.4 3.6 4.8 6.0 7.2 8.4 9.6 10.8 12

global radiation (w/m²)

PET, Priestley & Taylor (mmd⁻¹)

ET GLEAM (mmd⁻¹)

50 100 150 200 250 300 350 400

0.0 0.5 1.0 1.5 2.0 2.5 3.0 3.5 4.0 4.5 5.0

0.1 0.2 0.3 0.5 0.7 1 2 4 5 10 16

b)

1/32° 1/16° 1/8° 0.25°

0.5° 0.75° 1° 2°

% estimated aggregation bias in ET estimates

−200 −100 −50 −10 −5 0 5 10 35 50 100 200

c)

Figure S3. Same as in Fig. S2 but for another arbitrarily selected day (18.07.2004).

[Figure]

Figure S4. Standard deviation of altitude across several grid scales calculated from 500 m resolution topographic data (Bundesamt für Landestopographie, 1991). The spatial patterns of topographic variability at each grid scale are similar to spatial patterns of the median of daily aggregation biases shown in Fig 2 of the manuscript.

[Figure]

Figure S5. Soil moisture (m$^3$ m$^{-3}$) averaged over several grid scales calculated from 500 m resolution soil moisture data (m$^3$ m$^{-3}$) for a arbitrarily selected day (29.05.2004). The 500 m resolution soil moisture data are simulated by the PREVAH hydrological model (Brunner et al., 2019; Speich et al., 2015; Orth et al., 2015; Zappa et al., 2003).

[Figure]

Figure S6. Same as in Fig. S5 but for another arbitrarily selected day (21.07.2004).

**Data availability**

We will upload the source data for this study to a FAIR repository and provide the URL with the final version of the paper.